# Experimental Investigation of Isothermal Section in the La–Co–Ni System at 723 K

Kailin Huang, Liming Xiao, Qingkai Yang, Lu Yang, Zhuobin Li, Zhao Lu *, Qingrong Yao *, Jianqiu Deng ⓘ, Lichun Cheng, Caimin Huang, Qianxin Long, Jiang Wang ⓘ and Huaiying Zhou

Guangxi Key Laboratory of Information Materials, School of Material Science and Engineering, Guilin University of Electronic Technology, Guilin 530004, China
* Correspondence: luzhao_gx@163.com (Z.L.); qingry96@guet.edu.cn (Q.Y.)

**Abstract:** The isothermal section of the La–Co–Ni ternary system at 723 K has been constructed in this work by using X-ray diffraction (XRD), scanning electron microscopy, and energy dispersion spectroscopy techniques (SEM-EDS). The experimental results show no existence of ternary compounds at 723 K. The isothermal section consists of 16 two-phase regions and 8 three-phase regions. $La_3Co$ and $La_3Ni$, $La_2Co_3$ and $La_2Ni_3$, $La_2Co_7$ and $La_2Ni_7$, and $LaCo_5$ and $LaNi_5$ form a continuous solid solution. The ternary solid solubility of Ni in $LaCo_{13}$ phase and $La_2Co_{1.7}$ phase was determined to be 15.61 at.% and 9.61 at.%, respectively. The solid solubility of Co in the $LaNi_3$, $La_7Ni_3$, and $LaNi$ phases was 18.07 at.%, 5.62 at.%, and 8.49 at.%, respectively. The present experimental results are important for the design of $La(Fe,Si)_{13}$-based magnetic refrigeration materials.

**Keywords:** La–Co–Ni; phase diagram; X-ray diffraction; SEM observation





## 1. Introduction

Magnetic cooling is a novel, energy-efficient, and environmentally friendly technology that aims to replace conventional vapor-compression technology for air conditioning, space heating, and domestic refrigerators/freezers. It is based on the magnetocaloric effect (MCE), which occurs in magnetic materials, in particular those undergoing a magnetic or magneto-structural phase transition [1–3]. The $NaZn_{13}$-type $La(Fe,Si)_{13}$-based alloy system has attracted great attention among reported magnetic refrigerants, as it exhibits tunable giant entropy change and small hysteresis at ambient conditions and contains a minimal amount of rare-earth elements with high Fe-content and high magnetic moment [4,5]. However, its phase transition temperature is around 200 K, so it is difficult to cool it at room temperature. The introduction of transition element Ni can effectively improve the stability of $La(FeSi)_{13}$ compounds, and Co can increase its magnetic phase transition temperature [6–10]. By optimizing the Fe/Co/Ni/Si ratio of the $La(FeCoNiSi)_{13}$ alloy, the magnetic phase transition temperature of the alloy can be adjusted to room temperature, which makes the application of magnetic refrigeration technology at room temperature very possible. Phase diagrams are very important for studying the effects of the elements and are beneficial for alloys in design and fabrication [11–15]. Up to now, however, the current report on the isothermal section of the La–Co–Ni ternary system is the La < 32.2% isothermal section of the La–Co–Ni ternary phase diagram determined by Zheng et al. [16] in 1981, and the phase diagram of La–Co–Ni is not complete. Therefore, here we experimentally determined the isothermal cross-section of the La–Co–Ni ternary system at 723 K.

## 2. Literature Review

### 2.1. The La–Co System

The phase diagram of the Co–La system was first studied in detail by Velge [17], who measured the phase diagram of the entire compositional region by metallography, X-ray

diffraction, and thermal analysis. The phase formation of $La_3Co$, $La_2Co_{1.7}$, $La_2Co_3$, $La_2Co_7$, $LaCo_5$, and $LaCo_{13}$ was determined at 727 K, 843 K, 968 K, 1073 K, 1363 K, and 1458 K, respectively. After that, the phase diagram was re-investigated by Ray [18], and it was confirmed that $La_2Co_{1.7}$, $La_2Co_3$, $La_2Co_7$, $LaCo_5$, and $LaCo_{13}$ were formed by peritectic reaction and found one compound of the $La5Co19$, which [17] did not report, and the peritectic reaction of $La_5Co_{19}$ was confirmed at a temperature of about 1141 K. In 2006, Wang et al. [19] re-optimized the thermodynamics of the Co–La binary system, and the calculated data were consistent with the experimental results of Buschow and Velge [17]. The calculated phase diagram of the Co–La system is shown in Figure 1a. In 2019, Iwase et al. [20] reported the crystal structure of the compound $La_5Co_{19}$ in detail and found its good hydrogen storage properties.

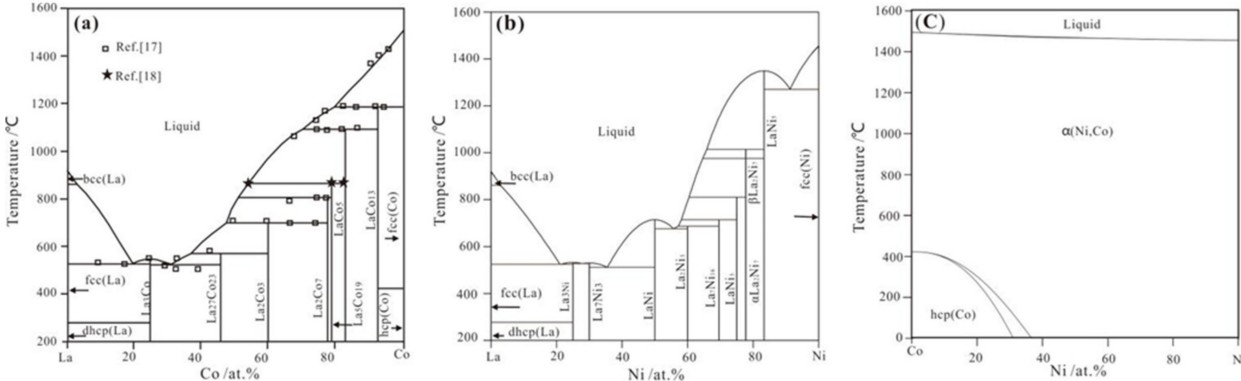

**Figure 1.** (**a**) The calculated phase diagram of the Co–La system with the experimental data [19]; (**b**) the calculated phase diagram of the La–Ni system [21]; (**c**) the calculated phase diagram of the Co–Ni system [22].

### 2.2. The La–Ni System

In 1972, Buschow and Mal [23] conducted a detailed study of the Ni–La (50–100 at% Ni) system using thermal analysis, metallography, and X-ray diffraction techniques, and they confirmed the stability of five phases $LaNi$, $LaNi_{1.4}$, $LaNi_2$, $LaNi_3$, $La_2Ni_7$, and $LaNi_5$. In 1991, Zhang et al. [24] re-determined the Ni–La binary system phase diagram (50–83.3 at% Ni) and found that the $LaNi_2$ phase could not exist stably, and the $LaNi_{2.286}$ phase was confirmed to be the $La_7Ni_{16}$ phase. In 2003, Zhou et al. [25] studied the phase diagram of the Ni–La–Si ternary system and found that there are eight intermediate compounds at the Ni–La end, namely: $La_3Ni$, $La_7Ni_3$, $LaNi$, $La_7Ni_{16}$, $La_2Ni_3$, $La_2Ni_7$, $LaNi_5$, and $LaNi_3$. In 1998, Du et al. [21] studied the phase diagram of the Ni–La binary system and found nine intermetallic compounds, namely $La_3Ni$, $La_7Ni_3$, $LaNi$, $La_7Ni_{16}$, $La_2Ni_3$, $La_2Ni_7$, $\beta La_2Ni_7$, $LaNi_5$, and $LaNi_3$. The calculated phase diagram of the Co–La system is shown in Figure 1b. In 2000, Dischinger and Schaller [26] used DSC technology to measure the heat capacity of all intermetallic compounds in the La–Ni system at 923~1023 K but did not consider the $La_5Ni_{19}$ phase. In addition, they reported a new phase ($La_4Ni_{17}$), but this conclusion was not confirmed by other researchers. In 2008, An et al. [27] re-studied the Ni–La binary system and discovered the $La_5Ni_{19}$ phase, which was formed by the peritectic reaction between the liquid phase and the $LaNi_5$ phase at a reaction temperature of 1276 K. In 2017, Subotenko et al. [28] found that $LaNi_2$ is stable at elevated temperatures and decomposes into the $La_7Ni_{16}$ phase below 1003.5 K.

### 2.3. The Co–Ni System

The Co–Ni system is a simple isomorphous system. It is completely mutually soluble in the whole concentration range of Co and Ni. In 2019, Zhou et al. [22] conducted a thermodynamic evaluation of the Co–Ni–Ta system, where the calculated phase diagram of the Co–Ni system was shown as Figure 1c.

### 2.4. The La–Co–Ni System

In 1981, a part of the ternary phase diagram of La–Co–Ni (La < 32.2% isothermal section at room temperature) was determined by Zheng [14]. Four single-phase regions were: $\alpha$Co, $\beta$Ni, $La_3Co$, and $LaCo_{5x}Ni_{5-5x}$; five two-phase regions were: $\alpha$Co + $\beta$Ni, $LaCo_{13}$ + $\alpha$Co, $LaCo_{13}$ + $\beta$Ni, $\beta$Ni + $LaCo_{5x}Ni_{5-5x}$, $LaCo_{13}$ + $LaCo_{5x}Ni_{5-5x}$; and two three-phase regions were: $\alpha$Co + $\beta$Ni + $LaCo_{13}$ and $LaCo_{13}$ + $\beta$Ni + $LaCo_{5x}Ni_{5-5x}$. Because the phase diagram of the La–Co–Ni system was not verified in detail due to the limitation of the early experimental conditions, this work is based on the study of the phase diagram of the La–Co–Ni system.

The known structures of the single and binary phases are listed in Table 1 [16,17,20,25].

**Table 1.** The data on the crystal structures of the compounds of the La–Ni, Ni–Co, and La–Co binary systems.

| Compound | Pearson Notation | Space Group | Structure Type | Lattice Parameters | | Reference |
|---|---|---|---|---|---|---|
| | | | | a (nm) | c (nm) | |
| $LaCo_{13}$ | F112 | Fm-3c | $NaZn_{13}$ | a = 11.334(1) | — | [17] |
| $LaCo_5$ | hP6 | P6/mmm | $CaCu_5$ | a = 5.100(5) | c = 3.968(5) | [17] |
| $La_5Co_{19}$ | hR72 | R-3m | $Ce_5Co_{19}$ | a = 5.130(5) | c = 49.50(4) | [20] |
| $La_2Co_7$ | hR54 | R-3m | $Gd_2Co_7$ | a = 5.101(5) | c = 24.511(4) | [17] |
| $La_2Co_3$ | 0S20 | Cnca | $La_2Ni_3$ | a = 10.34(1) | c = 7.811(5) | [17] |
| $La_2Co_{1.7}$ | mS8 | C2/m | $La_2Co_{1.7}$ | a = 8.4536(1) | c = 4.2723(2) | [17] |
| $La_3Co$ | Op16 | pnma | $Fe_3C$ | a = 7.277(9) | c = 6.575(8) | [17] |
| $\alpha$Co | Cf4 | Fm-3m | Cu | a = 3.5447 | — | [16] |
| $\beta$La | Cf4 | Fm-3m | Cu | a = 3.7740 | c = 12.171 | [16] |
| $La_7Ni_3$ | — | P63mc | $Fe_7Th_3$ | a = 1.0130 | c = 0.6462 | [25] |
| LaNi | — | Cmcm | BCr | a = 0.3907 | c = 0.4396 | [25] |
| $La_2Ni_3$ | 0S20 | Cmca | $La_2Ni_3$ | a = 0.5118 | c = 0.7907 | [25] |
| $La_7Ni_{16}$ | — | I42m | $La_7Ni_{16}$ | a = 0.7355 | c = 1.451 | [25] |
| $LaNi_3$ | — | R3m | $BeNb_3$ | a = 0.5086 | c = 2.501 | [25] |
| $La_2Ni_7$ | hR55 | P63/mmc | $Ce_2Ni_7$ | a = 0.5085 | c = 2.471 | [25] |
| $LaNi_5$ | — | P6/mmm | $CaCu_5$ | a = 0.5016 | c = 0.3983 | [25] |
| $\beta$Ni | Cf4 | Fm-3m | Cu | a = 3.5446 | — | [25] |

## 3. Experimental Procedure

The phase relationships of the La–Co–Ni ternary system were constructed by equilibrated alloys. High purity of La (99.99%), Co (99.99%), and Ni (99.99%) were used as raw materials. The alloys with the mass of 3 g were prepared by arc melting with a non-consumable tungsten electrode under the protection of a high-purity argon atmosphere. The samples were turned over and re-melted four times to improve alloy homogeneities. The melting loss was less than 1 mass.%. The alloys were annealed at 723 K for 1440 h. After that, the samples were taken from the muffle furnace and quenched quickly in ice water.

The phase consistence and crystal structure of the samples were identified by means of XRD (Rigaku D/max2550VB, Tokyo, Japan) using Cu K$\alpha$ radiation. The diffractometer was operated at 40 kV and 40 mA, and the 2$\theta$ scan ranges from 20° to 90° with a step size of 0.02° and a counting time of 5 s per step. The microstructural features and the elemental distribution of the alloy were characterized using SEM-EDS and XRD methods.

## 4. Results and Discussion

### 4.1. Microstructure

The twenty-two La–Co–Ni alloys annealed at 723 K for 1440 h were examined by SEM-EDS and XRD. It should be noted that the accuracy of the EDS results is about 0.01%. The experimental results of phase compositions measured by EDS and phase identified by XRD are presented in Table 2. Figure 2 shows SEM images and XRD patterns of the representative La–Co–Ni alloys annealed at 723 K for 1440 h. According to the SEM-EDS results and the XRD analysis, it can be seen in Figure 2a that alloy #5 ($La_{85}Co_5Ni_{10}$) was composed of $La_3(Co,Ni)$ and $\beta$La phases, which agrees with the results of XRD results. The light-gray phase and white phase in Figure 2a correspond to the $\beta$La and $La_3(Co,Ni)$ phases, respectively. Their

compositions were $La_{73.21}Co_{17.30}Ni_{9.49}$ and $La_{94.53}Co_{4.21}Ni_{1.21}$, respectively. So, the alloy $La_{85}Co_5Ni_{10}$ is confirmed to be located in the two-phase region, $La_3(Co,Ni) + \beta La$. The BSE image of the no. 6 ($La_{62}Co_6Ni_{32}$) sample is presented in Figure 2b, showing the co-existence of two-phase $La_7(Co,Ni)_3$ (white) + LaNi(gray). Their compositions are $La_{68.79}Co_{5.62}Ni_{25.59}$ and $La_{49.61}Co_{3.37}Ni_{47.02}$, respectively. The result is consistent with the XRD results. So, the alloy $La_{62}Co_6Ni_{32}$ is confirmed to be located in the two-phase region, $La_7Ni_3 + LaNi$. The BSE image of the no. 1 ($La_9Co_{85}Ni_6$) sample annealed at 723 K for 1440 h is presented in Figure 2c, showing the co-existence of the three-phase area $LaCo_5$(white) + $LaCo_{13}$(gray) + $\alpha Co$(black). The result is consistent with the XRD results. Figure 2d presents the BSE image of the no. 4 ($La_{68}Co_{26}Ni_6$) sample alloy annealed at 723 K for 1440 h. The white and gray phases are the $La_3(Co,Ni)$ phase and $La_2Co_{1.7}$ phase. The no. 18 ($La_{34}Co_{11}Ni_{55}$) sample displays the alloy that consists of the $La_2Ni_3$(white) and $LaNi_3$(gray) phase in Figure 2f. The XRD pattern identified the result, as can be seen in Figure 2e, and their compositions are $La_{40.22}Co_{3.63}Ni_{56.15}$, $La_{25.90}Co_{17.99}Ni_{56.11}$, and $La_{22.96}Co_{29.03}Ni_{48.01}$, respectively. Figure 2g,h shows BSE images and XRD patterns of the no. 12 ($La_8Co_{10}Ni_{82}$) sample annealed at 723 K for 1440 h. According to the SEM-EDS results and the XRD analysis, it can be seen in Figure 2h that alloy #12 ($La_8Co_{10}Ni_{82}$) was composed of $LaNi_5$ and $\beta Ni$ phases, which agrees with the results of XRD results, as can be seen in Figure 2g.

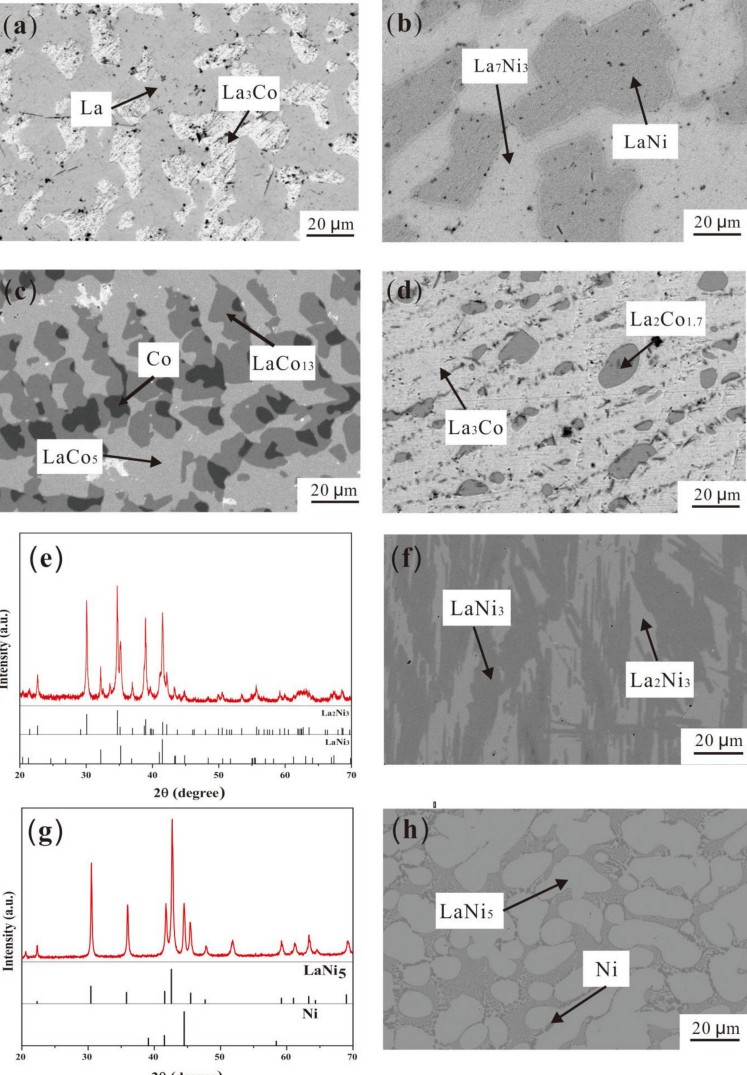

**Figure 2.** The XRD pattern and SEM micrograph for alloys annealed at 723 K for 60 days: (**a**) no. 5 ($La_{85}Co_5Ni_{10}$); (**b**) no. 6 ($La_{62}Co_6Ni_{32}$); (**c**) no. 1 ($La_9Co_{85}Ni_6$); (**d**) no. 4 ($La_{68}Co_{26}Ni_6$); (**e,f**) no. 18 ($La_{34}Co_{11}Ni_{55}$); (**g,h**) no. 12 ($La_8Co_{10}Ni_{82}$).

**Table 2.** Composition of the La–Co–Ni phases according to the EDS data.

| Alloy | Nominal Composition (at.%) | EDS Measurement | | | | XRD Results |
|---|---|---|---|---|---|---|
| | | La (at.%) | Co (at.%) | Ni (at.%) | Phase | |
| 1 | $La_9Co_{85}Ni_6$ | 16.06 | 72.30 | 11.65 | $LaCo_5$ | $LaCo_5$ |
| | | 0.01 | 95.31 | 4.68 | $\alpha Co$ | $\alpha Co$ |
| | | 6.92 | 77.47 | 15.61 | $LaCo_{13}$ | $LaCo_{13}$ |
| 2 | $La_7Co_{44}Ni_{49}$ | 15.41 | 72.13 | 12.46 | $LaCo_5$ | $LaCo_5$ |
| | | 0.40 | 95.03 | 4.56 | $\alpha Co$ | $\alpha Co$ |
| | | 0.50 | 14.20 | 85.30 | $\beta Ni$ | $\beta Ni$ |
| 3 | $La_{12}Co_{60}Ni_{28}$ | 0.01 | 94.31 | 5.68 | $\alpha Co$ | $\alpha Co$ |
| | | 16.06 | 71.3 | 12.65 | $LaCo_5$ | $LaCo_5$ |
| 4 | $La_{68}Co_{26}Ni_6$ | 53.79 | 43.07 | 3.13 | $La_2Co_{1.7}$ | $La_2Co_{1.7}$ |
| | | 72.54 | 16.76 | 10.70 | $La_3Co$ | $La_3Co$ |
| 5 | $La_{85}Co_5Ni_{10}$ | 73.21 | 17.30 | 9.49 | $La_3Co$ | $La_3Co$ |
| | | 94.53 | 4.21 | 1.26 | $\beta La$ | $\beta La$ |
| 6 | $La_{62}Co_6Ni_{32}$ | 68.79 | 5.62 | 25.59 | $La_7Ni_3$ | $La_7Ni_3$ |
| | | 49.61 | 3.37 | 47.02 | $LaNi$ | $LaNi$ |
| 7 | $La_{51}Co_{34}Ni_{15}$ | 40.53 | 45.79 | 13.68 | $La_2Co_3$ | $La_2Co_3$ |
| | | 49.73 | 8.49 | 41.78 | $LaNi$ | $LaNi$ |
| | | 52.73 | 37.66 | 9.61 | $La_2Co_{1.7}$ | $La_2Co_{1.7}$ |
| 8 | $La_{44}Co_{12}Ni_{44}$ | 40.35 | 17.96 | 41.69 | $La_2Ni_3$ | $La_2Ni_3$ |
| | | 50.67 | 8.49 | 40.84 | $LaNi$ | $LaNi$ |
| 9 | $La_{44}Co_{49}Ni_7$ | 53.15 | 41.63 | 5.22 | $La_2Co_{1.7}$ | $La_2Co_{1.7}$ |
| | | 41.53 | 50.08 | 8.39 | $La_2Co_3$ | $La_2Co_3$ |
| 10 | $La_{70}Co_5Ni_{25}$ | 70.41 | 5.45 | 24.14 | $La_7Ni_3$ | $La_7Ni_3$ |
| | | 72.54 | 10.76 | 16.70 | $La_3Ni$ | $La_3Ni$ |
| | | 49.45 | 1.28 | 49.26 | $LaNi$ | $LaNi$ |
| 11 | $La_{72}Co_5Ni_{23}$ | 70.41 | 5.06 | 24.53 | $La_7Ni_3$ | $La_7Ni_3$ |
| | | 72.86 | 5.62 | 21.53 | $La_3Ni$ | $La_3Ni$ |
| 12 | $La_8Co_{10}Ni_{82}$ | 17.51 | 25.20 | 57.29 | $LaNi_5$ | $LaNi_5$ |
| | | 0.05 | 5.37 | 94.58 | $\beta Ni$ | $\beta Ni$ |
| 13 | $La_8Co_{90}Ni_2$ | 7.41 | 77.26 | 15.33 | $LaCo_{13}$ | $LaCo_{13}$ |
| | | 0.40 | 95.03 | 4.56 | $\alpha Co$ | $\alpha Co$ |
| 14 | $La_{10}Co_{88}Ni_2$ | 16.06 | 70.30 | 13.66 | $LaCo_5$ | $LaCo_5$ |
| | | 7.41 | 77.26 | 15.33 | $LaCo_{13}$ | $LaCo_{13}$ |
| 15 | $La_{33}Co_{24}Ni_{43}$ | 40.12 | 21.01 | 38.87 | $La_2(Co,Ni)_3$ | $La_2(Co,Ni)_3$ |
| | | 22.96 | 29.03 | 48.01 | $La_2(Co,Ni)_7$ | $La_2(Co,Ni)_7$ |
| 16 | $La_{31}Co_{42}Ni_{27}$ | 39.80 | 41.12 | 19.08 | $La_2(Co,Ni)_3$ | $La_2(Co,Ni)_3$ |
| | | 22.83 | 45.98 | 31.19 | $La_2(Co,Ni)_7$ | $La_2(Co,Ni)_7$ |
| 17 | $La_{30}Co_{63}Ni_7$ | 39.12 | 56.13 | 7.75 | $La_2(Co,Ni)_3$ | $La_2(Co,Ni)_3$ |
| | | 22.67 | 69.83 | 7.50 | $La_2(Co,Ni)_7$ | $La_2(Co,Ni)_7$ |
| 18 | $La_{34}Co_{11}Ni_{55}$ | 40.20 | 10.10 | 49.70 | $La_2Ni_3$ | $La_2Ni_3$ |
| | | 25.52 | 18.07 | 56.41 | $LaNi_3$ | $LaNi_3$ |
| 19 | $La_{36}Co_3Ni_{61}$ | 40.22 | 3.63 | 56.15 | $La_2Ni_3$ | $La_2Ni_3$ |
| | | 30.50 | 2.82 | 66.68 | $La_7Ni_{16}$ | $La_7Ni_{16}$ |
| 20 | $La_{29}Co_4Ni_{67}$ | 30.49 | 4.98 | 64.53 | $La_7Ni_{16}$ | $La_7Ni_{16}$ |
| | | 25.90 | 17.99 | 56.11 | $LaNi_3$ | $LaNi_3$ |
| 21 | $La_{60}Co_{19}Ni_{21}$ | 74.93 | 6.62 | 18.45 | $La_3Ni$ | $La_3Ni$ |
| | | 50.12 | 6.71 | 43.17 | $LaNi$ | $LaNi$ |
| | | 53.94 | 36.45 | 9.61 | $La_2Co_{1.7}$ | $La_2Co_{1.7}$ |
| 22 | $La_{57}Co_{14}Ni_{29}$ | 74.98 | 6.67 | 18.35 | $La_3Ni$ | $La_3Ni$ |
| | | 50.67 | 8.49 | 40.84 | $LaNi$ | $LaNi$ |
| | | 53.43 | 37.67 | 8.90 | $La_2Co_{1.7}$ | $La_2Co_{1.7}$ |

### 4.2. Isothermal Section for La–Co–Ni at 723 K

Based on the experimental results obtained in this work and the information of relevant binary systems taken from the literature, the isothermal section of the La–Co–Ni system at 723 K in the whole concentration region was constructed and is shown in Figure 3. $La_3Co$, $La_2Co_{1.7}$, $La_2Co_3$, $La_2Co_7$, $LaCo_5$, $La_5Co_{19}$, $LaCo_{13}$, $La_3Ni$, $La_7Ni_3$, $LaNi$, $La_7Ni_{16}$, $La_2Ni_7$, $LaNi_5$, and $LaNi_3$ phases were found to be stable at 723 K. Among them, $La_3Co$ and $La_3Ni$, $La_2Co_3$ and $La_2Ni_3$, $La_2Co_7$ and $La_2Ni_7$, and $LaCo_5$ and $LaNi_5$ form continuous solid solutions. The ternary solid solubility of Ni in the $LaCo_{13}$ phase and $La_2Co_{1.7}$ phase is about 15.61 at.% and 9.61 at.%, respectively. In addition, the ternary solid solubility of Co in the $LaNi_3$ phase, $La_7Ni_3$ phase, and $LaNi$ phase are about 18.07 at.%, 5.62 at.%, and 8.49 at.%, respectively. No ternary compounds were found. Unfortunately, the phase

region in the dotted line cannot be determined because the relevant sample is not balanced. The system consists of 16 two-phase regions and 8 three-phase regions. In contrast to the room-temperature isothermal cross-section [28], in the region of La < 32.2%, βNi in the single-phase region of αCo can be solubilized at room temperature, and the cross-section at this temperature is αCo + βNi + La(Co,Ni)$_5$ in the three-phase region. There is the LaCo$_{13}$ + βNi + LaCo$_5$ three-phase region at room temperature, which transforms the LaCo$_{13}$ + αCo + LaCo$_5$ three-phase region at 723 K. In addition, the two-phase region αCo + βNi and the three-phase region αCo + βNi + LaCo$_{13}$ disappeared at 723 K and were replaced by the two-phase region αCo + La(Co,Ni)$_5$. Furthermore, the solid solution limit of βNi in LaCo$_{13}$ is found to be 15.61 at.% by means of the no. 1 sample.

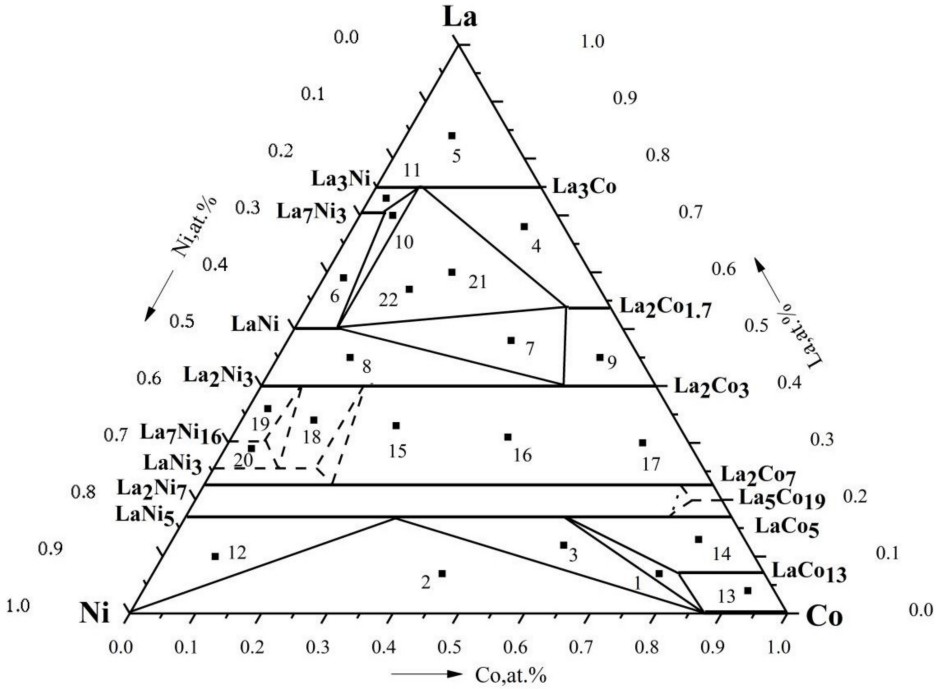

**Figure 3.** Isothermal section of the La–Co–Ni system at 723 K (in the whole concentration region).

The eight three-phase regions are: La$_3$Ni + La$_7$Ni$_3$ + LaNi, La$_7$Ni$_{16}$ + La$_2$Ni$_3$ + LaNi$_3$, LaNi + La$_2$Co$_3$ + La$_2$Co$_{1.7}$, LaNi + La$_3$Ni + La$_2$Co$_{1.7}$, αCo + βNi + La(Co,Ni)$_5$, La$_2$Ni$_3$ + LaNi$_3$ + La$_2$Ni$_7$, LaCo$_{13}$ + αCo + LaCo$_5$, and La$_2$Co$_7$ + La$_5$Co$_{19}$ + LaCo$_5$. The 16 two-phase regions are: La$_3$(Co,Ni)$_5$ + βLa, La$_3$Ni + La$_7$Ni$_3$, La$_7$Ni$_3$ + LaNi, La$_2$Co$_3$ + La$_2$Co$_{1.7}$, La$_3$Co + La$_2$Co$_{1.7}$, LaNi + La$_3$Ni + La$_2$Co$_{1.7}$, La$_2$Ni$_3$ + LaNi, LaCo$_{13}$ + LaCo$_5$, La$_2$Co$_7$ + La$_2$Co$_3$, LaCo$_{13}$ + αCo, LaCo$_5$ + αCo, βNi + LaNi$_5$, La$_2$Ni$_3$ + La$_7$Ni$_{16}$, LaNi$_3$ + La$_7$Ni$_{16}$, La$_2$Ni$_3$ + La$_2$Ni$_7$, and LaNi$_3$ + La$_2$Ni$_3$.

## 5. Conclusions

In this work, phase equilibria in the La–Co–Ni system at 723 K were measured using equilibrated alloy samples. Eight three-phase regions and sixteen two-phase regions were determined in the isothermal sections of the La–Co–Ni ternary system. La$_3$Co and La$_3$Ni, La$_2$Co$_3$ and La$_2$Ni$_3$, La$_2$Co$_7$ and La$_2$Ni$_7$, and LaCo$_5$ and LaNi$_5$ form a continuous solid solution. The ternary solid solubility of Ni in the LaCo$_{13}$ phase and La$_2$Co$_{1.7}$ phase is about 15.61 at.% and 9.61 at.%, respectively. In addition, the ternary solid solubility of Co in the LaNi$_3$ phase, La$_7$Ni$_3$ phase, and LaNi phase is about 18.07 at.%, 5.62 at.%, and 8.49 at.%, respectively. No ternary compounds were found.

**Author Contributions:** Data curation, K.H.; formal analysis, K.H., L.X., Q.Y. (Qingkai Yang), L.Y., Z.L. (Zhuobin Li), Z.L. (Zhao Lu) and Q.Y. (Qingrong Yao); investigation, K.H. and L.X.; methodology, K.H., L.X., Q.Y. (Qingkai Yang), L.Y. and Z.L. (Zhuobin Li); software, K.H.; writing—original draft preparation, K.H.; writing—review and editing, Z.L. (Zhao Lu); project administration, Z.L. (Zhao Lu) and Q.Y. (Qingrong Yao); Contribution to the design, Z.L. (Zhao Lu), Q.Y. (Qingrong Yao), J.D., L.C., C.H., Q.L., J.W. and H.Z.; interpretation of data, J.D., L.C., C.H., Q.L., J.W. and H.Z. All authors have read and agreed to the published version of the manuscript.

**Funding:** This work was supported by the National Natural Science Foundation of China (grant nos.: 52061007 and 52261004), the Science and Technology Project of Guangxi (grant nos.: AB21220028, AA22068084, AA18242023, and AD21220018), the Natural Science Foundation of Guangxi (grant no.: 2021GXNSFDA075009), Guangxi Key Laboratory of Information Materials (grant nos.:191012-Z and 211034-Z), basic scientific research of young teachers in Guangxi universities (grant no.: 2021KY0198), and the innovation program for University students (grant no.: 201910595037) are acknowledged.

**Institutional Review Board Statement:** Not applicable.

**Informed Consent Statement:** Not applicable.

**Data Availability Statement:** Not applicable.

**Conflicts of Interest:** The authors declare no conflict of interest.

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
