# Peer review of "Experimental Investigation of Isothermal Section in the La–Co–Ni System at 723 K"

_metals, doi:10.3390/met12101747_

Round 1

Reviewer 1 Report

This paper is an experimental investigation of the equilibrium phases present at 723 K for the ternary La-Co-Ni system.

1. In the introduction, refs 9-12 are cited regarding the importance of phase diagrams. But of these papers, only the latter is relevant. All the others refer to some specific system.

2. Is there any information about the accuracy of the chemical compositions given in Table 2? Nothing is said.

3. Fig. 2 gives just the nominal composition of the experimental alloys. This could be completed with the tie-lines as they are measured (Table 2), in the same or in another diagram.

4. Fig. 2: some lines are drawn as dotted lines; but the meaning of this is not even mentioned in the text.

5. Line 23: inparticula // in particular

6. Line 50: “Ref” should be a number.

7. Line 136: beseen // be seen

8. References: please delete the redundant reference number in refs. 1-14, 18-21 and 24-26.

9. Refs. 14-26: a [J] is included. Please delete.

10. Ref. 16, line 219: please review paper title (Cobalt).

11. Line 211: 1000 ?C // 1000 °C

Author Response

Dear Ms. Karolina Prevedei, dear Reviewers,

Thank you very much for the e-mail concerning our manuscript entitled “Experimental investigation of isothermal sections at 723 K in the La–Co–Ni systems”. We really appreciate your instructive advice and thoughtful comments. Enclosed please find the revised manuscript. All the hints and remarks have been carefully checked and considered in our revised manuscript. All the modifications in our revised manuscript are highlighted in red color. The following describes how these comments are addressed item by item in the revised manuscript.

Reviewer #1:

Comment 1: In the introduction, refs 9-12 are cited regarding the importance of phase diagrams. But of these papers, only the latter is relevant. All the others refer to some specific system.

Response: Thank you so much for your comment. We have adopted your suggestion. We only kept refs 9 and 10 in our manuscript.

9.Fartushna I , Mardani M , Khvan A , et al. Experimental Investigation of Fe–Co–La System: Liquidus and Solidus Projections[J]. Springer US, 2020(4).

10.Fartushna, I., Mardani, M., Khvan, A., Cheverikin, V., & Kondratiev, A. (2020). Experimental investigation of phase equilibria in the Co–Fe–La system at 600 and 500 °C. Calphad, 70, 101794.

Comment 2: Is there any information about the accuracy of the chemical compositions given in Table 2? Nothing is said.

Response: Firstly, The loss of all the samples was less than 0.001% after melting. Therefore, the chemical compositions of the prepared sample can be ensure equal to nominal composition; Secondary, the accuracy of EDS result is about 0.01%, and we have add it in the revised manuscript.

Comment 3: Fig. 2 gives just the nominal composition of the experimental alloys. This could be completed with the tie-lines as they are measured (Table 2), in the same or in another diagram.

Response : Thank you so much for your comment. We have adopted your suggestion.

Comment 4: Fig. 2: some lines are drawn as dotted lines; but the meaning of this is not even mentioned in the text.

Response : Thank you very much for your careful review. We have given the meaning of the dotted lines in the Fig. 2 in revised manuscript。

Comment 5: Line 23: inparticula // in particular

Response : We have adopted your suggestion.

Comment 6:Line 50: “Ref” should be a number.

Response : We have adopted your suggestion.

Comment 7: Line 136: beseen // be seen

Response : We have adopted your suggestion.

Comment 8: References: please delete the redundant reference number in refs. 1-14, 18-21 and 24-26.

Response : We have adopted your suggestion.

Comment 9: Refs. 14-26: a [J] is included. Please delete.

Response 9: We have adopted your suggestion.

Comment 10: Ref. 16, line 219: please review paper title (Cobalt).

Response : We have adopted your suggestion.We re-cite the relevant literature.[13] Chen X , Chen Y G , Tang Y B , et al. Effect of Ce,Co,B on formation of LaCo_(13)-structure phase in La(Fe,Si)_(13) alloys. Transactions of Nonferrous Metals Society of China , 2014,24(03):705-711.

Comment 11: Line 211: 1000 ?C // 1000 °C

Response: We have adopted your suggestion.

Reviewer 2 Report

The comments were sent in a separate file.

Author Response

Dear Ms. Karolina Prevedei, dear Reviewers,

Thank you very much for the e-mail concerning our manuscript entitled “Experimental investigation of isothermal sections at 723 K in the La–Co–Ni systems”. We really appreciate your instructive advice and thoughtful comments. Enclosed please find the revised manuscript. All the hints and remarks have been carefully checked and considered in our revised manuscript. All the modifications in our revised manuscript are highlighted in red color. The following describes how these comments are addressed item by item in the revised manuscript.

Reviewer #2:

Comments to the Authors.

The submitted manuscript entitled “Experimental Investigation of Isothermal Section in the 2 La– Co–Ni System at 723 K” (metals-1939002) is in the scope of Metals journal. In present manuscript, the isothermal section of the La-Co-Ni ternary system at 723 K has been determined using different experimental methods: XRD, SEM and EDS. The aim of this work focused on construction of ternary phase equilibrium diagram to design the La(Fe,Si)13- based magnetic refrigeration materials. Phase diagrams are very important to study the element doping effects and beneficial for alloys in design and fabrication. At the beginning the authors discovered the absence of ternary compounds at the isothermal section which consists of 16 two-phase regions and 8 three-phase regions. The formation of continuous solid solution of La3Co and La3Ni, La2Co3 and La2Ni3, La2Co7 and La2Ni7, LaCo5 and LaNi5 was observed.Additionally, the considerable ternary solid solubility of Ni in LaCo13 phase and La2Co1.7 phase, and the solid solubility of Co in LaNi3, La7Ni3, and LaNi phases were also identified.

But I have additional remarks which (I believe) can improve the manuscript text:

Some of this remarks, especially, concerning tables composition are addressed to Editor, as well.

Response: Thanks for your comment! We have adopted your suggestion and we have carefully cheeked the language of our manuscript.

  1. Page 1, line 23: …in particulart, I guess.

Response:Thank you so much for your comment.We have adopted your suggestion.

  1. Page , line: 27: … magnetic moment[4-5]. – space is necessary.

Response2:Thank you so much for your comment.We have adopted your suggestion.

  1. Page 1, line: 28-29: However, its phase transition temperature is around 200 K, so it is difficult to cool it at room temperature – could you be so kind and explain what kind of phase transition is at 200 K. Magnetic? Ferro -> para? It should be emphasised that Co has the highest Curie temperature (TC) of elements.

Response:Thank you so much for your comments.We have adopted your suggestion.

At 200K, the first-stage phase change occurs. The magnetic entropy change of LaFe11.8Si1.2 compound is approximately into a platform in the temperature range of 20 K (about 182-205 K) near TC. As the platform width increases with x, at x = 1.8, the platform completely disappears and the magnetic entropy change is λ-shaped, which is a typical second-order phase change feature. We show that the LaFe13-xSix compounds with low Si content exhibit a first-order magnetic phase transition, with a magnetic field-induced IEM transition from paramagnetic to ferromagnetic forces above the Curie temperature. LaCo13 is the only stable compound with a Curie temperature TC of about 1318 K.

  1. Could you also explain (in one sentence) why Ni was used to improve stability of

La(FeSi)13 compound. Its TC is rather low about 627 K.

Response:Thank you so much for your comment. The effect of Ni substitution on the magnetocaloric properties of La(Fe,Si)13 compounds. Sample quality has been optimized by a combination of induction melting and suction casting techniques, which allowed to shorten the annealing time by an order of magnitude and expand the existing range of LaFe11.6-xNixSi1.4 phase with cubic NaZn13-type structure up to x =0.4. Magnetocaloric studies show that the first order phase transition was transformed into a second order type with increasing Ni concentration.

  1. Page 1, line 36: … control of the compound …. What do you want to control? It seems that you can control the compound (phase) stability?

Response5:Thank you so much for your comment. We have corrected.

  1. Page 1, line 40: …ternary system at 723 K is investigated – was investigated?

Response6:Thank you so much for your comment. Here we experimentally determined the isothermal cross section of the La-Co-Ni ternary system at 723 K.

  1. Page 2, line 46-47: Could you explain what do you mean by “The phase stabilities of La3Co, La2Co1.7, La2Co3, La2Co7, LaCo5 and LaCo13 were determined at 727 K, 843 K, 968 K, 1073 K, 1363 K and 1458 K”. The stability was observed at these temperature sequence or there are the temperatures of congruent or incongruent melting? It can be a little uncomfortable to understand for readers.

Response7:Thank you so much for your comment. We express Phase formation of La3Co La2Co1.7 La2Co3 La2Co7 LaCo5 and LaCo13 was determined at 727 K 843 K 968 K 1073 K 1363 K and 1458 K, respectively.

  1. Page 2, line 49: …by peritectic reaction… - is it one the same reaction?

Response8:Thank you so much for your comment, it is one the same reaction. 

At 1185℃, the LaCo13 phase is formed by peritectic reaction between the liquid phase and the fcc(Co) phase.

At 1090℃, the LaCo5 phase is formed by peritectic reaction between the liquid phase and the LaCo13 phase.

At 800℃, the La2Co7 phase is formed by peritectic reaction between the liquid phase and the LaCo5 phase.

At 695℃, the La2Co3 phase is formed by peritectic reaction between the liquid phase and the La2Co7 phase.

At 570℃, the La2Co1.7 phase is formed by peritectic reaction between the liquid phase and the La2Co3 phase.

  1. Page 2, line 50: …and La5Co19 was found, which Ref did not report… - what do you mean by Ref? This all sentence is little bit not clear. Please improve it. Especially, that in next sentence the information of “The temperature is about 1141 K” is also confusing. Which peritectic reaction (I guess) have you on mind?

Response9:Thank you so much for your comment.We re-express it as:"and found one compound of the La5Co19, which Ref[15] did not report, and the peritectic reaction of La5Co19 was confirmed, at a temperature of about 1141 K.

10.Page 2, line 52: … it is found that the peritectic reaction temperature... The exact peritectic reaction should be revealed. Also the quantity of experimental temperatures of Ray and Buschov (not only Velge) should be enclosed. How can you compare these temperatures if “Ray [16] did not give …. the exact temperature of the peritectic reaction.

Response10:Thank you so much for your comment. We modify the relevant content.

  1. Page 2, line 54: ….did not give details… - did not give the details…

 Response11:Thank you so much for your comment. We have adopted your suggestion.

12.Page 2, line 57: …Buschow and Velge [15] et al. - What do you mean by et al. in this sentence?

Response12:Thank you so much for your comment. We modify the relevant content.

  1. Page 2, line 61: …Buschow et al. [19]… - Buschow and Mal [19].

Response13:Thank you so much for your comment. We have adopted your suggestion.

  1. Page 2, line 62: Please decide which notation to use: X-ray (as previously) or x-ray.

Response14:Thank you so much for your comment. We have adopted your suggestion.

  1. Page 2, line 63: There is a wrong sentence order. Better may be: they confirmed the stability of five phases (or compounds) LaNi, LaNi1.4, LaNi2, LaNi3, La2Ni7, and LaNi5.

Response15:Thank you so much for your comment. We have adopted your suggestion.

  1. Page 2, line 69: …Dischinger and Schaller [22] and others… - what do you mean by others? In references there are only two authors? The same line: … the thermodynamic data… - which thermodynamic data?

Response16:Thank you so much for your comment. We modify the relevant content.“Dischinger and Schaller[22] used DSC technology to measure the heat capacity of all intermetallic compounds in the La-Ni system at 923~1023 K, but did not consider the La5Ni19 phase.” 

  1. Page 2, line 71: … a new composite phase… - what the authors understand by composite phase in this system?

Response17:Thank you so much for your comment. We modify the relevant content. “a new phase”

  1. Page 2, line 72: In 1998, Du et al. [23] studied the … - As I understand the author’s intention it is not chronologically presented. The year 2000 is in line 68.

Response18:Thank you so much for your comment. We modify the relevant sentences.

  1. Page 2, lines 776-78: …found that LaNi2 belongs to the high temperature phase and decomposes into La7Ni16 phase below 1003.5 K. It is not strict. Could you explain how the LaNi2 phase can decompose into La7Ni16? And LaNi2 is stable at elevated temperatures (not belongs). Please, consider of the additional presentation of phase diagrams of assessed binary systems.

Response19:Thank you so much for your comment. We modify the relevant content.LaNi2 is stable at elevated temperatures.

At 1103.5K, the La7Ni16 phase is formed by peritectic reaction between the liquid phase and the LaNi2 phase.

  1. Could you be so kind and inform at the beginning of binary systems description that the exact information on phases are in Table 1? in which the binary phase diagram of co-ni was shown as 1

Response20:We have adopted your suggestion.

  1. Page 2, lines 80-81: Could you improve the sentence: Co-Ni has(?) only α phase in the whole concentration range and is completely solid solution in solid and liquid state. It is not the proper English.

Response21:We have adopted your suggestion.

It is completely mutually soluble in the whole concentration range of Co and Ni.

  1. Page 2, lines 83: … which confirmed this conclusion… - You should be careful in such opinions because during system evaluation (or better assessment), generally, it is common to make assumptions of phase stabilities basing on experimental data.

Response22: We modify the relevant content.

  1. Page 2, line 85: It seems that it was only a part of phase diagram.

Response23: We have adopted your suggestion.

  1. Page 2, line 86: Could you reveal the experimental methods of Zheng et al. [14] determination of isothermal section at room temperature?

Response24:Thank you so much for your comment. Alloys were prepared with a Al2O3 crucible in a high-frequency induction electric furnace in a high-purity argon atmosphere. All alloys were sealed in a vacuum quartz tube at 900℃ for ten days, and then cooled to room temperature at 10℃/ h.

  1. Page 2, lines 87-88: Diacritical marks (“;” and “:”) should be written without spaces.

Response25: We have adopted your suggestion.

  1. Table 1: It seems to be better to move Lattice parameters column after crystal structure information (Pearson number, Space group and structure type).

 Response26:We have adopted your suggestion.

  1. Page 3, line 99: I suggest to use the mass instead of weight. Mass is more basic.Pge

Response27:We have adopted your suggestion.

  1. Page 3, line 101: the Saxon Genitive in alloy’s is not necessary. … alloy homogenities.

Response29: We have adopted your suggestion.

  1. Page 3, line 102: better to use: 1 mass%.

Response29: We have adopted your suggestion.

  1. Page 3, line 103: in an ice water.

Response30:We have adopted your suggestion.

  1. What was the argon gas purity? Because the lanthanum has very high affinity for oxygen. Have you observed any traces of oxides?

Response31:We used a high-purity argon gas(99.99at.%)

32.Page 3, line 104: The phase consistent …. I guess that you mean: consistence?

Response32: We have adopted your suggestion.

  1. Page 3, line 113: …SEM-EDS and XRD methods.

Response33:We have adopted your suggestion.

  1. Page 3, line 117: …La3(Co,Ni) and βLa phases…

Response34:We have adopted your suggestion.

  1. Page 3, line 118-119: …. βLa and La3(Co,Ni) phases, respectively.

Response35:We have adopted your suggestion.

  1. Page 3, line 121: The BSE image of the La62Co6Ni32 alloy… - please add the number of this sample from Table 2. It is not necessary to repeat the conditions of sample preparation. It was describe it experimental method description part.

Response36: We have adopted your suggestion.

  1. Page 3, line 122: … showing the co-existence of two-phase La7Ni3(white) + LaNi(gray). In ternary system you determined the solubility of Co in binary compounds. Therefore, please, consider other notation, for example La7(Co,Ni)3 etc.

 Response37: We have adopted your suggestion.

  1. Page 4, line 127: …three-phase… It should be: three phases of three-phase area. It is not clear why the authors started to present their result from sample 5.

Response38: We have adopted your suggestion.Start with the number 5 from the rich La end.

  1. Page 4, line 128: … presents the BSE image of the…

Response39: We have adopted your suggestion.

  1. Page 4, line 129: The white and gray phases are… Again the preparation conditions

are needless here.

Response40: We have adopted your suggestion.

  1. Page 4, line 130: …displays the phase consists of… - the phase can not consist with 2

other phases.

Response41:We convert“ displays the alloy consists of ”.

  1. Page 4, line 131: … result,as can… - space before as is necessary.

Response42: We have adopted your suggestion.

  1. Page 4, line 133: … La22.96Co29.03Ni48.01,respectively. – again the space is necessary.

Response43: We have adopted your suggestion.

  1. Why the samples notation in text, in Figure caption 1 and in Table 2 are not consistent. In Table the subscripts for element contents were used.

Response45:We have adopted your suggestion, all adopt a subscript.

  1. Page 4, line 134: … for 1440 h .According… - … for 1440 h. According

Response45:We have adopted your suggestion.

  1. Page 4, line 134: …βNi phase,which agree with the results of XRD results. as can beseen..

Response46: We have adopted your suggestion.

  1. Page 6, line 152-153: … form continuous solid solutions. - It should be emphasized that the ranges of these solubilities are limited to several percent! It is not complete substitution.

Response47: We modify the relevant sentences.

  1. Page 7, line 157: … cross-section [26],In the … - space is necessary.

Response48We have adopted your suggestion.

  1. Page 7, line 160: …region.There is… - space is necessary. And “there is” or rather

“There are”. Or better: There is LaCo13+βNi+LaCo5 three-phase region…

Response49:We have adopted your suggestion.

  1. Page 7, line 161: …transforms to LaCo13+αCo+LaCo5 three-phase region at 723K.

Response50: We have adopted your suggestion.

  1. Page 7, line 164: …limit of βNi in LaCo13 is found to be 15.61 at.% at by means of

LaCo13+αCo+LaCo5. This sentence is not clear.

Response5 We modify the relevant sentences.

  1. Page 7, lines 165-171: The redundant spaces should be removed!

Response52We have adopted your suggestion.

  1. Fig.2.: Could you increase the sample point marks and use other marks: circles or squares)?

Response53:We have adopted your suggestion.

  1. References: line 202: … Synthesis and properties of NaZn 13 -type … - remove redundant spaces, please.

Response54 We have adopted your suggestion.

  1. References: line 205: 10M303 – as a number of pages.

Response55: We have adopted your suggestion.

  1.  
  2. References: line 208: … L.B. Liu,Experimental - space is necessary.

Response56: We have adopted your suggestion.

  1. References: line 211: …Nb–Co Ternary System at 1000 ?C, Mater. Des., 2017, 115, p

170-178 - ℃.

Response57: We have adopted your suggestion.

  1. References: line 212: … J.Phase Equilib. Diffus., 2014, 35(6), … Use the same style in all references with or without the issue number.

Response58: We have adopted your suggestion.

  1. References: line 214: [13]Q.R. – insert space.

Response59:We re-cite the relevant literature.[13] Chen X , Chen Y G , Tang Y B , et al. Effect of Ce,Co,B on formation of LaCo_(13)-structure phase in La(Fe,Si)_(13) alloys. Transactions of Nonferrous Metals Society of China , 2014,24(03):705-711.

  1. References: line 215: The tiltle should be: Crystal structure and phase relations of Pr2Fe14B-La2Fe14B system.

Response60:We re-cite the relevant literature.[15] Chen X , Chen Y G , Tang Y B , et al. Effect of Ce,Co,B on formation of LaCo_(13)-structure phase in La(Fe,Si)_(15) alloys. Transactions of Nonferrous Metals Society of China , 2014,24(03):705-711.

  1. References: line 215: Journal of Physics, 1982,19(5) : 674-679. But it should be: Acta Phys. Sin., 1982, 31(5): 674-679. doi: 10.7498/aps.31.674

Response61: We have adopted your suggestion.

  1. References: line 217: Buschow K, Velge W., Phase relations and… - add comma. The same in line 219.

Response62: We have adopted your suggestion.

  1. References: [16]: Ray, A. E. A Review of the Binary Rare Earth-Cobalt Alloy Systems. MAGNETISM AND MAGNETIC MATERIALS — 1973: Nineteenth An. Conf. (1974), doi:10.1063/1.2947224.

Response64:We have adopted your suggestion.

  1. References: [19]: Journal of The Less-Common… - the.

Response64:We have adopted your suggestion.

  1. References: [20]: 467-480 – but it should be 45-53. doi: 10.1016/0022-5088(91)90234- U

Response65: We have adopted your suggestion.

  1. References: [22]: 312: 201–210. – should be: 312(1-2): 201–210

Response66:We have adopted your suggestion.

  1. References: [23]: Alloys and Compounds, 1998, 264(1): - it shoul be: 264(1-2):

Response67: We have adopted your suggestion.

Reviewer 3 Report

I do not have any remarks.

Author Response

Dear Ms. Karolina Prevedei, dear Reviewers,

Thank you very much for the e-mail concerning our manuscript entitled “Experimental investigation of isothermal sections at 723 K in the La–Co–Ni systems”. We really appreciate your instructive advice and thoughtful comments. Enclosed please find the revised manuscript. All the hints and remarks have been carefully checked and considered in our revised manuscript. All the modifications in our revised manuscript are highlighted in red color. The following describes how these comments are addressed item by item in the revised manuscript.

Reviewer #3:

General Comments:

The manuscript provides a useful description of the experimental investigation of isothermal section in the Lanthanum–Cobalt–Nickel system at 723 K. The experimental results show no existence of ternary compounds at investigated temperature. According to authors the isothermal section consists of sixteen two-phase regions and eight three-phase regions. Results showed that La3Co and La3Ni, La2Co3 and La2Ni3, La2Co7 and La2Ni7, LaCo5 and LaNi5 only form continuous solid solutions. The ternary solid solubility of Ni in LaCo13 phase and La2Co1.7 phase were determined. Also the solid solubility of Co in LaNi3, La7Ni3, and LaNi phases were investigated. The strength of the study is an in-depth analysis of the issue and a thorough analysis of all phases occurring in this system at the tested temperature. The present experimental results are important for the design of magnetic refrigeration materials. The experimental work is very well-organized and clearly written. Therefore the reviewer thinks it is a valuable work and deserves for publishing in MDPI Metals.

Specific Comments:

The manuscript was written very correctly. The content contained is interesting and well prepared therefore I do not have any remarks.

Response: Thank you so much for your nice comments.
